# PlrA (MSMEG_5223) is an essential polar growth regulator in *Mycobacterium smegmatis*

**Samantha Y. Quintanilla, Neda Habibi Arejan, Parthvi B. Patel, Cara C. Boutte** *

Department of Biology, University of Texas Arlington, Arlington, TX, United States of America

* cara.boutte@uta.edu

## Abstract

Mycobacteria expand their cell walls at the cell poles in a manner that is not well described at the molecular level. In this study, we identify a new polar factor, PlrA, that is involved in restricting peptidoglycan metabolism to the cell poles in *Mycobacterium smegmatis*. We establish that only the N-terminal membrane domain of PlrA is essential. We show that depletion of *plrA* pheno-copies depletion of polar growth factor Wag31, and that PlrA is involved in regulating the Wag31 polar foci.

**Data Availability Statement:** All relevant data are within the paper and its Supporting Information files.

**Funding:** This work was funded by grant R15GM131317 from the National Institutes of

## Introduction

Expansion of the cell wall is critical for bacterial growth. In rod shaped bacteria, cells expand by elongating the rod, and then divide centrally to propagate daughter cells. Elongation occurs along the lateral walls in many proteobacterial and firmicute species [1]. Polar elongation occurs in several alphaproteobacterial species [2] and in Actinomyecetes [3, 4]. In the alpha-proteobacterium *Agrobacterium tumefaciens*, polar growth is dependent on Growth Pole Ring (GPR) protein, which forms a ring around the pole and is required for restricting peptidoglycan synthesis to the pole [5, 6]. Many Actinobacteria, including mycobacteria, also elongate at the poles [3, 7]. Actinobacterial polar growth is dependent on DivIVA-like proteins [8–12], which, like the GPR, restrict peptidoglycan synthesis to the poles [12]. The molecular mechanisms by which GPR and DivIVA proteins mediate polar elongation have not been described.

In Mycobacteria, the polar DivIVA-like protein is called Wag31. Wag31 localizes to the cell poles, with more Wag31 associated with the faster-growing old pole [9, 13, 14]. While it is clear that Wag31 is essential for establishing the pole and restricting peptidoglycan metabolism to the pole [12, 15], it is not at all clear how it works. Wag31 has no enzymatic domains and is cytoplasmic. In firmicutes, DivIVA proteins have been shown to recruit and activate other proteins involved in cell wall synthesis and regulation [16–20]. It is presumed that Wag31 somehow regulates polar peptidoglycan synthesis enzymes. However, despite being immuno-precipitated to find interaction partners in several studies [21–23], Wag31 has never been shown to interact with any polar peptidoglycan synthesis enzymes.

The complex of cytoplasmic, transmembrane and periplasmic regulators and cell wall enzymes that collectively mediate the ordered elongation of the cell wall is called the elonga-some. In lateral growers, the elongasome comprises cytoplasmic regulators like MreB, and peptidoglycan synthases including RodA and PBP2 [1, 24]. These proteins all function

Health to CCB. https://www.nih.gov The funders
had no role in study design, data collection and
analysis, decision to publish, or preparation of the
manuscript.

**Competing interests:** The authors have declared
that no competing interests exist.

together as a complex to allow the ordered insertion of new peptidoglycan. Wag31 has been
called an elongasome protein in Mycobacteria [4]; however, it is not at all clear that Mycobac-
terial elongation is mediated by a large protein complex that functions similarly to the elonga-
some characterized in *E. coli* and other lateral growers. First, recent work shows that the
critical peptidoglycan synthases required for polar growth are not even localized to the pole,
but instead are distributed nearly evenly around the cell membrane [25]. There must therefore
be a system to activate these proteins only near the pole. One model is that cell wall synthesis is
activated by the availability of cell wall precursors such as lipidII. Cell wall precursor enzymes
are localized largely to the Intracellular Membrane Domain, a biochemically distinct region of
the inner membrane that is localized mostly peri-polarly [15, 26, 27]. IMD enzymes, such an
MurG are therefore near the pole, but not at the pole, and they do not co-localize with Wag31.
Thus, it remains an open question how Wag31 can regulate the activity of enzymes when it
does not co-localize either with those enzymes or the production of their substrates.

Because polar growth in mycobacteria is so poorly understood, we reasoned that there are
likely many genes involved in this process that have not yet been characterized. In this study,
we describe initial characterization of one of those factors. In a previous study, we immuno-
precipitated the transmembrane division factor FtsQ from *Mycobacterium smegmatis* (*Msmeg*)
and identified several uncharacterized interactors [28]. One of these was MSMEG_5223
(Rv1111), which we found localized to the cell poles as well as the septum [28]. In this study we
show that MSMEG_5223, hereafter called PlrA, is essential for polar elongation in *Msmeg*, and
that, like Wag31, it restricts peptidoglycan metabolism to the pole. We also show that only the
N-terminus of PlrA is essential. Finally, we show that depletion of PlrA causes dysregulation of
the size of Wag31-mRFP polar foci, suggesting that PlrA may regulate Wag31 oligomerization
or membrane association.

## Results

### PlrA is essential for polar peptidoglycan metabolism and elongation

*plrA* (MSMEG_5223, Rv1111) is predicted by TnSeq to be essential for survival in both *Myco-
bacterium tuberculosis* [29] and *Msmeg* [30]. To study its function genetically, we made a
*Msmeg* strain, Ptet:: *plrA*, in which it can be transcriptionally depleted by removing the inducer
anhydrotetracycline (Atc). We confirmed that *plrA* expression in this strain allowed a normal
growth rate, and growth is like wild-type with a range of different levels of PlrA induction (Fig
1A, inset). We grew the Ptet:: *plrA* strain to logarithmic phase, then washed out the Atc and
measured survival by the colony forming unit assay. In these assays, we found that *plrA* deple-
tion only started to affect *Msmeg* growth and survival after ~24 hours of growth with depletion;
therefore, to observe effects, we diluted both the depletion and control cultures during growth
so that they did not reach stationary phase before the effect was observed. Our results show
that PlrA is essential for survival (Fig 1A). In this experiment, the lack of apparent growth in
the PlrA-induced sample is due to the dilutions, indicated with arrowheads. We then exam-
ined the *plrA* -depleted cells microscopically and found that after 30 hours of depletion, they
were short with bulgy poles (Fig 1B). These data show that *Msmeg* is unable to elongate prop-
erly and is unable to control cell wall structure at the poles without PlrA. We conclude that
PlrA is an essential polar elongation factor and name it *plrA* for pole regulator A.

*Mycobacteria* insert new peptidoglycan and other cell wall materials near the cell poles to
elongate [7, 13]. We used the fluorescent D-alanine HADA [31] to probe how the distribution
of peptidoglycan metabolism in the cells was affected by *plrA* depletion. We found that *plrA*
depletion led to HADA staining all along the lateral walls, instead of the typical polar and sep-
tal pattern (Fig 1B and 1C). This indicates that PlrA is required to restrict peptidoglycan

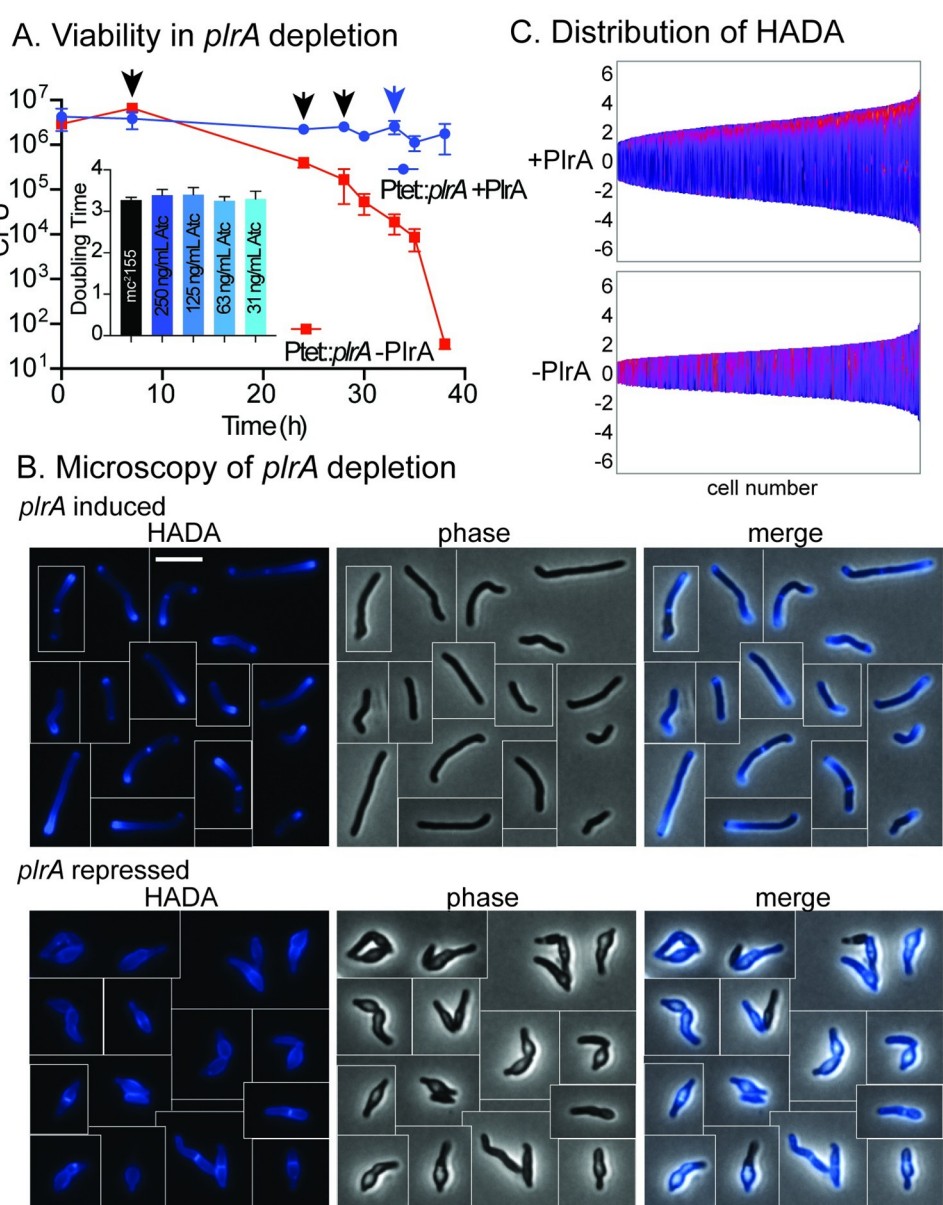

**Fig 1. PlrA is an essential polar growth regulator.** A) CFU of the Ptet::*plrA* strain in the presence (*plrA* induced) or absence (*plrA* repressed) of inducer Atc. Three independent replicate cultures were used for each condition. Black arrows represent when dilutions were performed in both cultures to prevent them reaching stationary phase. Blue arrow indicates when dilution was performed in the +Atc (+PlrA) culture only. Inset: Doubling time calculations from growth curves of the *Msmeg* mc²155 wild-type strain (black) and the Ptet::*plrA* strain induced with the indicated concentrations of Atc (shades of blue). A one-way ANOVA test with multiple comparisons shows that there are no significant differences between any of the presented doubling time measurements. B) Micrographs of a Ptet::*plrA* strain with *plrA* induced (top, +Atc) or repressed (bottom, -Atc), then stained with the fluorescent D-alanine HADA for 15 minutes. Cells from different images were cut and pasted together so that a representative collection of cells could be shown. The scale bar on the top left image is 5 microns, and applies to all images. C) Demographs of fluorescent images from the same experiment as cells in panel B. HADA intensity is indicated by color, with bright colors representative of high intensity, and dark color representing lower intensity. Intensity along the length of cells (Y axis of demograph) in *plrA* induced (top) and depleted (bottom) cells. Cells were sorted by size (X axis of demographs) and pole sorted, so that brightest pole is set at the top.

metabolism to the poles. Recent work shows that the HADA signal comes both from the insertion of new peptidoglycan [27], and from peptidoglycan remodeling by LD-transpeptidases [32] so these data cannot tell us whether, in the *plrA* depletion, new peptidoglycan synthesis is occurring all along the lateral walls, or whether peptidoglycan remodeling is just de-regulated. Because *plrA* depletion clearly leads to decreased cell length (Fig 1B), which is likely a result of decreased new cell wall material being inserted at the poles, we conclude that *plrA* may promote polar insertion of new peptidoglycan as well as polar-adjacent peptidoglycan remodeling. Short, bulgy cells and delocalized peptidoglycan metabolism, as measured by fluorescent D-amino acid staining, is also seen when the essential DivIVA homolog Wag31 is depleted [9, 12].

## PlrA localizes to the tips of both cell poles

In our previous study, we showed that PlrA localizes to cell poles and septa in *Msmeg* [28]. The polar growth regulator Wag31 has a similar localization pattern, and is seen to localize more strongly to the faster-growing old pole [23]. We stained cells expressing PlrA-GFPmut3 with HADA, which stains the old pole more brightly [32], in order to see if PlrA localizes in a similar pattern. We found (Fig 2) that PlrA-GFPmut3 had slightly brighter signal at the pole with brighter HADA staining, indicating that it localizes more to the faster growing pole. However, on average PlrA localization is similar between the two poles, compared to the significant difference in HADA staining between the new and old poles (Fig 2B and 2C). These data show that PlrA has a similar localization pattern as Wag31, in that it localizes to the pole tips [9]; however, the minimal asymmetry in PlrA localization suggests that the amount of PlrA at the pole is likely not responsible for regulating the asymmetry of polar elongation [13, 33]. We also observe PlrA-GFPmut3 signal at midcell locations which appear to be septal sites [28].

## The C-terminus of PlrA is dispensable, while the N-terminus is essential

PlrA has an N-terminal membrane domain with four predicted transmembrane passes and a C-terminal predicted cytoplasmic domain [34]. Because PlrA has no significant sequence similarity to any gene that has previously been experimentally characterized in bacteria, we sought to dissect its essentiality, by determining whether both or only one of these domains was essential. First, we used Consurf [35] to identify the relative conservation of each amino acid in the *Msmeg* protein. This analysis shows that the N-terminal membrane domain is more highly conserved than the C-terminal cytoplasmic domain (Fig 3A).

Then, we used L5 allele swapping [36] to attempt to replace the full-length *plrA* with *plrA*ΔCT (residues 1–117) or with *plrA*ΔNT (residues 118–368). In this method, the only copy of *plrA* on the genome is in a vector at the L5 site. By transforming in another L5-integrating vector with a different antibiotic resistance marker and a different allele of *plrA*, we could exchange alleles because the L5 integrase enzyme can excise the original L5 vector. Sometimes two vectors can be integrated at the L5 site, but we set up this experiment in such a way that transcription of *plrA* would be repressed in that case by the tetR repressor on the original L5 vector, so no colonies would form because *plrA* is essential (Fig 1A). So, we only expect colonies if the *plrA* allele on the second L5 vector can support growth (see Materials and Methods for more details). We found that a strain carrying only *plrA*ΔCT is viable, while a strain carrying only *plrA*ΔNT is not viable. This indicates that the more highly conserved N-terminal domain is essential, while the C-terminal domain is not (Fig 3B). All *plrA* alleles were cloned with a C-terminal strep tag, and we tested the stability of the PlrA truncations by western blot (Fig 3C). We made merodiploid strains of all the constructs so we could test whether the PlrAΔNT protein is stable. We found that PlrAΔNT is even more stable than the full length

## A. Micrographs of PlrA-GFPmut3

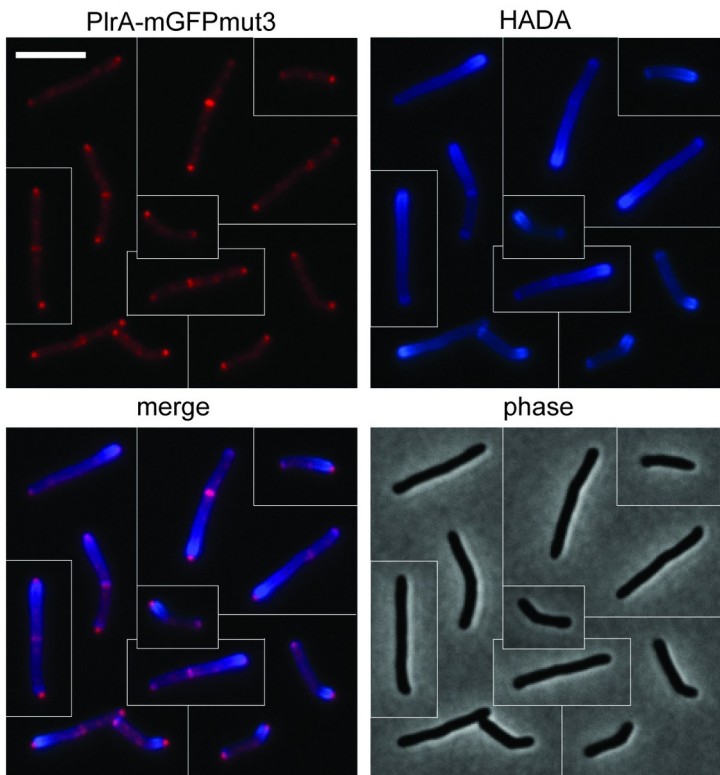

## B. Distribution of PlrA-GFPmut3

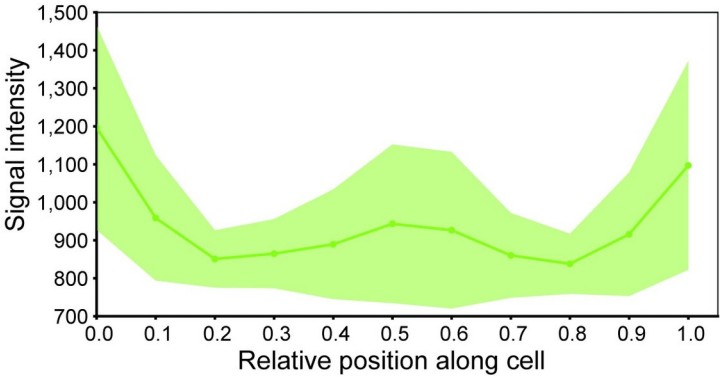

## C. Distribution of HADA staining

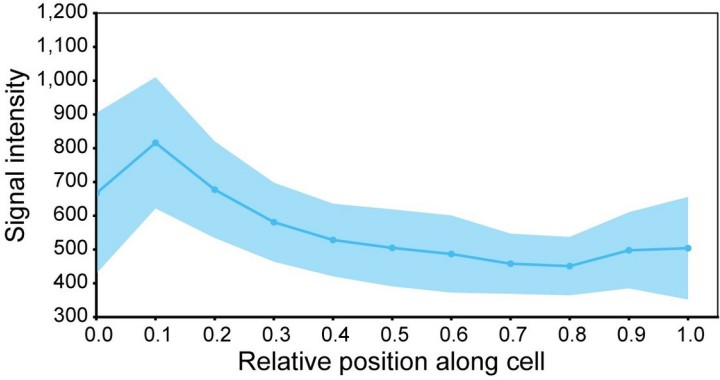

**Fig 2. PlrA localizes to the pole tips.** A) Micrographs of *Msmeg* mc²155 expressing PlrA-GFPmut3 as a merodiploid and stained with the fluorescent D-alanine HADA. The GFP signal was false colored to red to make the channels easier to distinguish. The scale bar on the top left image is 5 microns, and it applies to all images. Cells from different images were cut and pasted together so that a representative collection of cells could be shown. BC) Mean intensity of PlrA-GFPmut3 (B) and HADA (C) signal along the length of ~300 cells, at least 100 from each of three biological replicates. Center line is the mean signal, lighter area is the standard deviation. Cells were pole sorted, so the brightest pole in the HADA channel is set to 0, and the dimmer pole is set to 1 on the X axis.

protein, while the PlrAΔCT protein is less stable, and did not yield a detectable band on the western blot in either the merodiploid or the allele swap strain. This shows that the C-terminal domain of PlrA is not essential for function, and is not required merely for protein

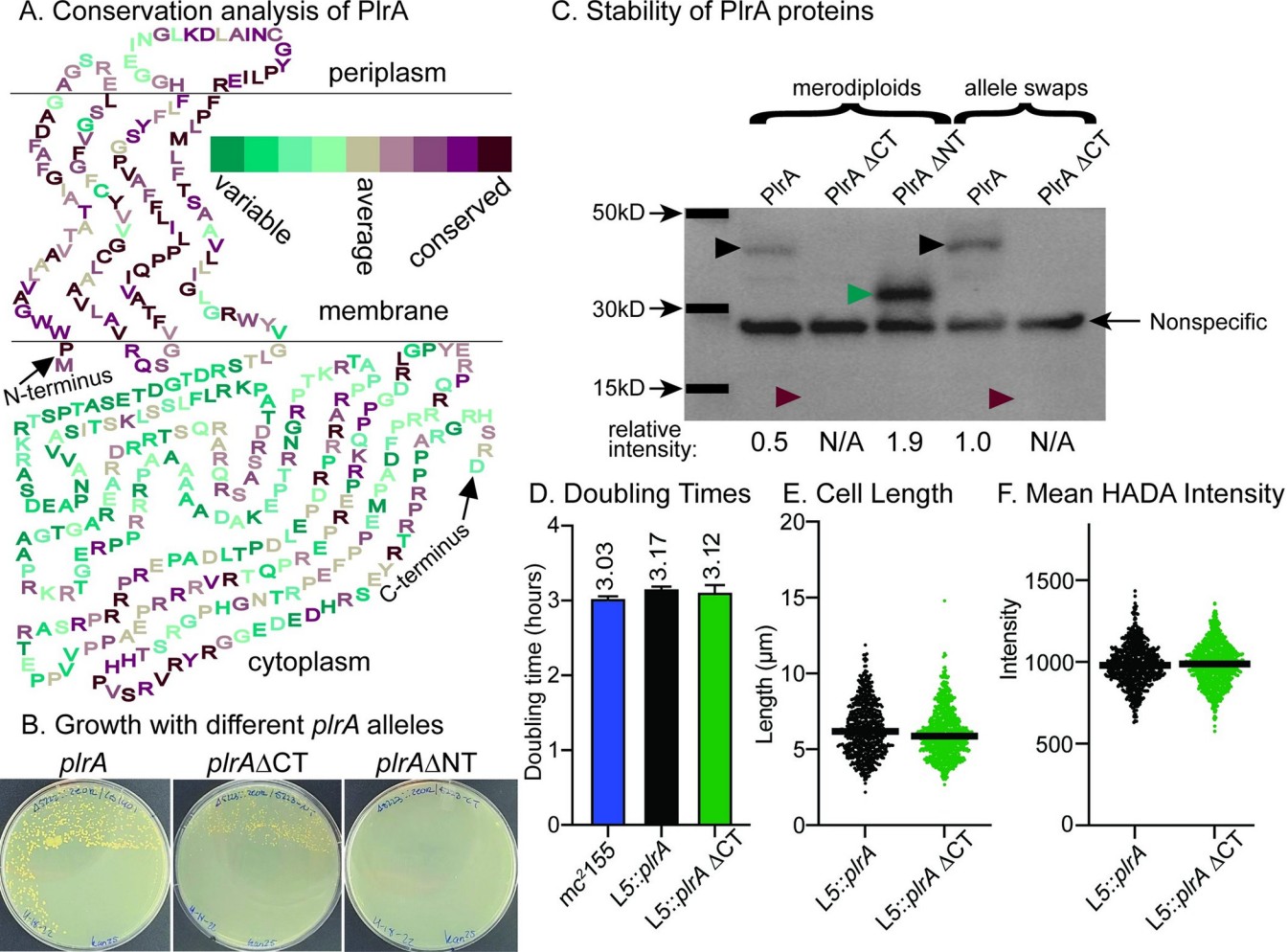

**Fig 3. The C-terminal domain of PlrA is dispensable.** A) PlrA protein sequence arranged in either periplasmic, membrane or cytoplasm as predicted by TMHMM. Amino acids are colored according to conservation as measured by Consurf analysis. B) Plates resulting from L5 allele swapping of the wild-type *plrA* with full-length *plrA*-strep, or *plrA*ΔCT-strep or *plrA*ΔNT-strep. The experiment was arranged so that the original, wild-type allele is lost in any surviving colonies–see Material and Methods for details. C) Western blot of *Msmeg* strains carrying either *plrA*-strep, *plrA*ΔCT-strep and *plrA*ΔNT-strep as merodiploids or allele swaps. PlrA-strep is 43kD and is indicated with black arrowheads, PlrAΔCT-strep is 13kD, and its expected location is indicated with maroon arrowheads, PlrAΔNT strep is 31kD, and is indicated with a teal arrowhead. Densitometry of the PlrA-strep bands was used to calculate the relative band intensities of each band compared to the WT PlrA-strep allele swap–shown at bottom of gel image. N/A indicates that the band was not apparent. D) Doubling times calculated from growth curves in 7H9 of the mc²155 parent, Δ*plrA* L5::*plrA*-strep, and Δ*plrA* L5::*plrA*ΔCT-strep strains. E) Cell lengths of the Δ*plrA* L5::*plrA*-strep, and Δ*plrA* L5::*plrA*ΔCT-strep strains in log. phase, as quantified from phase microscopy images by MicrobeJ analysis. F) Average HADA intensity per cell of cells from E.

stabilization. These data also suggest that very little PlrA is needed for survival, as the PlrAΔCT protein is undetectable by western, despite supporting growth. The conclusion that only a very small amount of PlrA protein–even below the limit of detection of a western blot–is required to support growth is corroborated by our experiments with the Ptet::*plrA* depletion strain (Fig 1A), where we did not observe growth defects in *plrA* depletion until after 24 hours of depletion with multiple dilutions, which is an estimated eight cell doublings, each of which would reduce the existing PlrA level by half.

We next tested whether the PlrA C-terminal domain contributes to growth in logarithmic phase. We found that the *plrAΔCT* strain had no defects in growth rate (Fig 3D), cell morphology (Fig 3E) or peptidoglycan metabolism as measured by fluorescent D-amino acid staining (Fig 3F). These data show that the C-terminal domain of PlrA is entirely dispensable for normal logarithmic phase growth.

## Depletion of PlrA causes atypical accumulation of Wag31 at the poles

Because the *plrA* depletion (Fig 1) exhibited a similar phenotype as the *wag31* depletion [12], we hypothesized that these two proteins may work together to regulate polar growth. We sought to determine whether Wag31 localization is dependent on PlrA. We transformed a vector expressing a Wag31-mRFP fusion into the Ptet::*plrA* depletion strain. We grew the resulting strain with or without the Atc inducer, then HADA-stained the cells and examined them microscopically. We found that Wag31-mRFP still localized to the cell poles in the cells depleted for *plrA* (Fig 4A). However, we observed that the size and intensity of the Wag31 foci was more variable in the *plrA*-depleted cells (Fig 4B–4D). Many of the *plrA*-depleted cells, especially the shorter cells which are presumably more severely depleted, had unusually bright and large foci, while other cells had very dim Wag31-mRFP foci (Fig 4B). In the control cells (left side of Fig 4), HADA and Wag31-RFP intensity were greater at the same cell pole in each cell, which we expect to be the old pole [32]. In the *plrA*-depleted cells, the new pole can be identified in V-snapping cells as the pole at the vertex of the V. We found, in these V-snaps, that the old pole was often dimmer by HADA than the new pole, while the new pole is usually the one bulging. We found that the unusually bright Wag31 foci were often at HADA-dim old poles (Fig 4A and 4B), and therefore the cell pole that was brighter by HADA was not also brighter by Wag31-RFP (Fig 4C).

To probe the relationship between peptidoglycan metabolism—as measured by HADA staining—and Wag31-mRFP localization, we plotted the maximum values of fluorescence intensity at each cell pole against each other (Fig 4D). We find that in the control cells, the presumed old poles (brighter by HADA) had a roughly gaussian distributions of both Wag31 and HADA signal across the population, and there is not a significant correlation between the signal in these two channels. This suggests that in the control cells, all the old poles are similar with respect to peptidoglycan metabolism and Wag31, which is what we expect since all old poles grow at the same rate [13, 37]. There was a weak correlation between Wag31-RFP signal and HADA signal in control cells at the presumed new poles, however (Fig 4D). This makes sense as the new pole undergoes changes throughout the cell cycle: right after division it does not elongate, and so we see less peptidoglycan metabolism (Fig 4B and 4D), but as the cells mature, the new pole becomes elongation-competent [37], and we saw a corresponding increase in Wag31-mRFP and HADA signal (Fig 4B and 4D). In the *plrA*-depleted cells, we saw a loss of Wag31-RFP signal intensity clustering in both poles, and the correlation between HADA signal and Wag31-mRFP signal at the HADA-dim pole was lost. These data suggest that PlrA helps control the size or structure of the Wag31 focus, as well as the polarity of peptidoglycan metabolism.

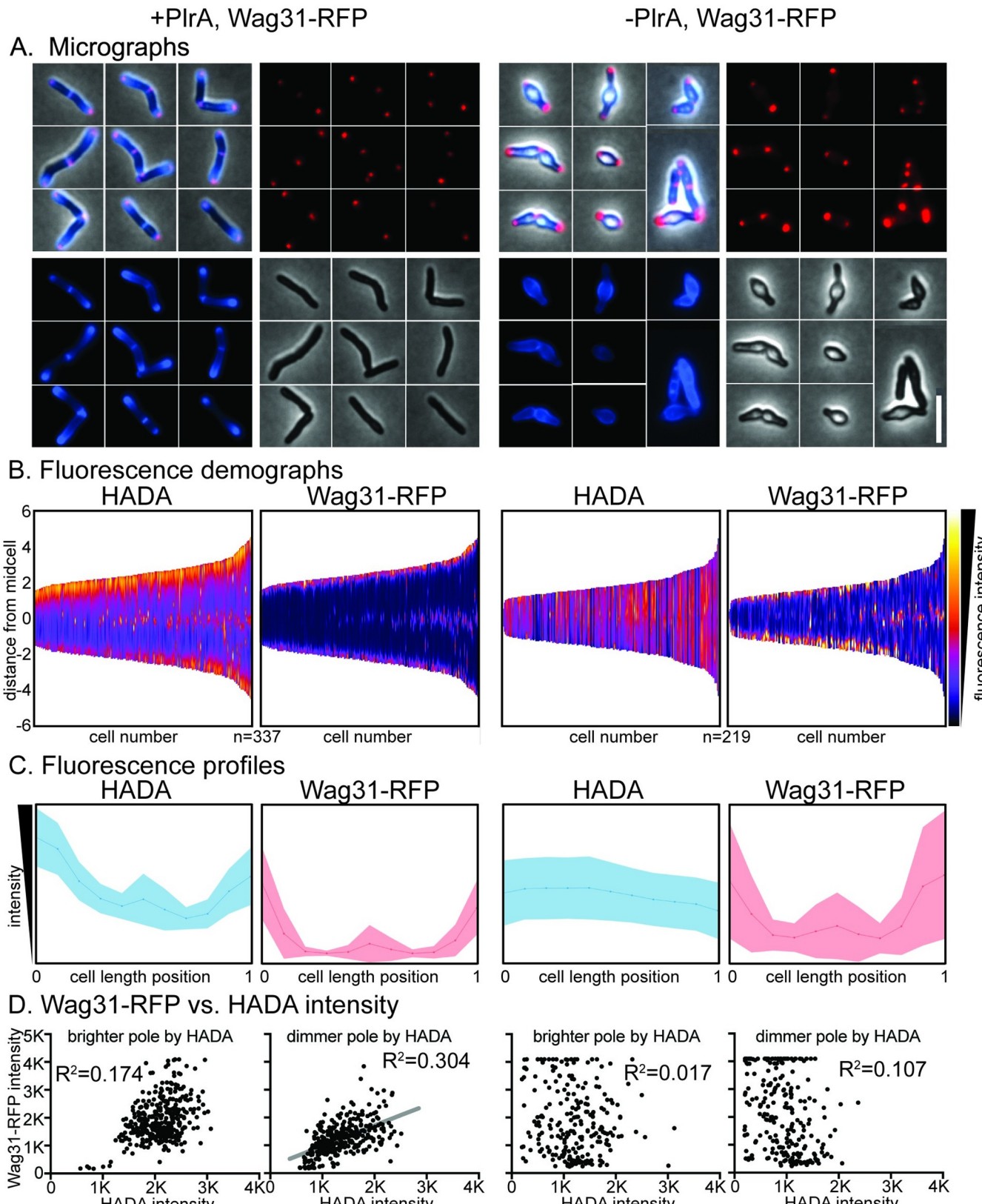

**Fig 4. PlrA helps regulate the Wag31 polar foci.** A) Micrographs of Ptet::*plrA* Wag31-mRFP strains induced (left) and depleted (right) for *plrA*, and stained with HADA. Blue = HADA fluorescence image. Red = Wag31-RFP fluorescence image. Scale bar on bottom right is 5 microns and applies to all images. B) Demographs of fluorescence intensity of the cell populations imaged in (A). *plrA* induced cells are on the left, *plrA* depleted cells on the right. Cells are arranged shortest to longest along the X axis, and arranged so the pole with the brighter HADA signal is positioned at the top. C) Mean fluorescence intensities (Yaxis) of all the cells from (A,B) at 11 points along the length of each cell. *plrA* induced cells are on the left, *plrA* depleted cells on the right. Darker line in the center is the mean, and shaded area is the standard deviation. Cells are sorted so that the pole with the brighter HADA intensity is set to 0 on the X axis. Both HADA graphs have the same intensity values between 0–2400 on the Y axis. Both RFP graphs have the same intensity values between 0–3200 on the Y axis. D) Maximum Wag31-RFP signal (Y axis) plotted against the maximum HADA signal (X axis) at each cell pole. *plrA* induced cells are on the left, *plrA* depleted cells on the right. $R^2$ values were calculated by linear regression analysis. The gray line is the linear fit on the only graph with a correlation.

## Discussion

It remains an open question how much the "elongasome" model from lateral growing bacteria should serve as inspiration for developing models for polar growth in Actinomycetes and Alphaproteobacteria. In this model, cytoplasmic regulators help control periplasmic cell wall enzymes through trans-membrane protein interactions. In both the alphaproteobacterium *A. tumefaciens* and the actinobacterium *C. glutamicum*, this model seems to hold up to some extent, as the key polar peptidoglycan transglycosylase enzymes are localized at the cell poles [38, 39], and in *C. glutamicum*, is anchored there through a cytoplasmic regulator [39]. We do wish to note that mis-localization of proteins can occur easily during over-expression, for example [40, 41]. The model of the elongasome complex does not seem to apply in *Mycobacterium smegmatis*, where the putative polar growth regulator Wag31 does not co-localize with either peptidoglycan precursor enzymes or transglycosylases [15, 23, 25–27]. Since polar growth appears to work so differently in Mycobacteria, we reasoned that there must be other essential factors involved in this process, and that characterizing those factors may help establish a new model for Mycobacterial polar growth.

Our work shows that the N-terminal membrane domain of PlrA is the domain essential for polar growth, and the cytoplasmic domain appears to not have any function during logarithmic phase growth (Fig 3). Non-enyzmatic membrane proteins involved in cell growth and division can either have roles regulating enzymes in the periplasm [42] or the cytoplasm [43], or they can bind and regulate other factors through their membrane-pass regions [44, 45]. The fully functional PlrAΔCT protein has only a four-amino acid cytoplasmic loop, while there is a 22-amino acid periplasmic loop. The most highly conserved residues are in the membrane passes and in a region of the periplasmic loop. We therefore think it most likely that PlrA regulates either a periplasmic enzyme, or another membrane protein through membrane contacts.

What could PlrA be doing to Wag31? Previous work has shown that increased Wag31 at sites in the cell causes increases in polar growth [46]. Our work shows that a large Wag31 focus can be inactive in polar growth, when PlrA is missing. The asymmetry of Wag31 foci at the poles, which correlates with the asymmetry of growth, suggests that the conformation or size of the homo-oligomeric Wag31 network could be involved in regulating polar growth (Fig 4B and 4D). Our work suggests that PlrA is required for a bright Wag31 focus to permit polar peptidoglycan synthesis (Fig 4). Perhaps PlrA helps control the chemical structure or shape of the pole, which may, in turn, affect Wag31 oligomer organization. Depletion data suggests that pole structure is dependent on PlrA, not solely on the presence of the Wag31 oligomer, as the Wag31 oligomer remains in place when the poles bulge due to plrA depletion (Fig 4A).

## Materials and methods

### Bacterial strains and culture conditions

*M. smegmatis* mc$^2$155 was cultured in 7H9 (Becton, Dickinson and Co, Sparks, MD) medium with additives as described [28] or plated on LB Lennox agar. E. coli DH5a, TOP10, or

XL1-Blue cells were used for cloning. The antibiotic concentrations used for *M. smegmatis* were: 25 μg/ml kanamycin, 50 μg/ml hygromycin, 20 μg/ml nourseothricin, and 20 μg/ml zeocin. The antibiotic concentrations used for *E. coli* strains were: 50 μg/ml kanamycin, 100 μg/ml hygromycin, 50 μg/ml zeocin, and 40 μg/ml nourseothricin. Anhydrotetracyline (Atc) was used at between 50 and 250 ng/ml for gene induction or repression.

## Strain construction

Knockout of *plrA* was made by first integrating a copy of *plrA* at the L5 site [47] in the nourseothricin-resistant pMC1s vector with a tet-inducible P750 promoter. The endogenous copies of the gene was then knocked out using double stranded recombineering, as described [48]. The knockout construct was made by PCR-stitching ~500 bp of DNA from directly upstream and downstream of the *plrA* gene to either side of a zeocin-resistance cassette. The double stranded DNA fragment was transformed into the mc$^2$155 L5::pMC1s- *plrA* / pNitRecET, and the resulting colonies were checked by PCR on either side of the knockout locus to confirm knockout. Vectors were assembled using Gibson cloning [49], some with the SSB enhancement [50]. After constructions and sequence confirmation, vectors were transformed into electro-competent *M. smegmatis* mc$^2$155, or into CB966 (see detailed plasmid, vector and strain information in the S1 File).

For the L5 allele swapping, a copy of *plrA* under the control of a tet-inducible promoter was cloned into a nuoR L5 vector carrying the TetR repressor and inserted into the *Msmeg* genome at the L5 phage integrase site, then the endogenous copy of *plrA* was deleted. We then cloned the full-length and truncation alleles of *plrA*, also under tet-promoters, into a kanR L5 integrating vector without the *tetR* gene. Transformation of these kanR vectors into the *Msmeg* strain carrying *plrA* only at the L5 site could result in either nuoR +kanR double integrants, or in kanR nuoS allele swaps. Because only the original L5 vector carries *tetR*, which will repress expression of either of the *plrA* alleles in either L5 vector, we plate the transformants without the Atc inducer and therefore select against the double integrants. Because *plrA* is essential (Fig 1A), in this setup, we will only get colonies on the transformation plate if the *plrA* allele in the second, kanR vector is functional enough to support growth.

## Colony forming unit assay

Clones of the Ptet:: *plrA* strain were grown to logarithmic phase in 7H9 with nourseothricin, zeocin, and 500 ng/mL of anhydrotetracyline (Atc). All cultures were washed to remove Atc, and diluted to OD = 0.1, Atc was added to half the cultures and allowed to grow. At the 7 hour time point, both cultures were diluted to OD = 0.01. At the 24 hour time point, both cultures were diluted to OD = 0.2. At the 28 hour time point, both cultures were diluted to OD = 0.1. At the 35 hour time point only the +Atc culture was diluted to OD = 0.01. Atc was re-added to the +Atc cultures only during the dilutions. CFU were measured on LB plates with nourseothricin, zeocin and Atc.

## Microscopy and image analysis

Microscopy was performed on living cells immobilized on Hdb-agarose pads. A Nikon Ti-2 widefield epifluorescence microscope with a Photometrics Prime 95B camera and a Plan Apo 100x, 1.45 NA objective was used for imaging. The images of the strains expressing fusions to the GFPmut3 fluorescent protein were taken with a 470/40nm excitation filter, a 525/50nm emission filter and a 495nm dichroic mirror. The HADA images were taken using a 350/50nm excitation filter, a 460/50nm emission filter and a 400nm dichroic mirror. The mRFP images were taken with a 560/40nm excitation filter, a 630/70nm emission filter and a 585nm dichroic

mirror. All images were processed using NIS Elements software and analyzed using FIJI and MicrobeJ [51].

## Western blotting

*Msmeg* strains carrying different versions of PlrA-strep were grown to log. phase, lysed by bead beating, extracted by laemmli buffer, then spun, and the supernatants were separated SDS-PAGE. Proteins were transferred to PVDF membranes, blocked with 5% milk in TTBS, then probed with either α-strep (1:1000, Abcam, ab76949) or α-RpoB (1:1000, Thermo Fisher Scientific, MA1-25425), and then goat anti-rabbit IgG HRP-conjugated secondary antibody (1:10,000, Thermo Fisher Scientific 31460).

## Supporting information

**S1 File.**
(DOCX)

**S1 Raw image.**
(PDF)

## Author Contributions

**Conceptualization:** Samantha Y. Quintanilla, Cara C. Boutte.

**Formal analysis:** Samantha Y. Quintanilla, Neda Habibi Arejan, Parthvi B. Patel, Cara C. Boutte.

**Investigation:** Samantha Y. Quintanilla.

**Supervision:** Cara C. Boutte.

**Writing – original draft:** Cara C. Boutte.

**Writing – review & editing:** Samantha Y. Quintanilla, Neda Habibi Arejan, Cara C. Boutte.

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
