## [Decision Letter · Decision Letter 0]

2 Nov 2022

PONE-D-22-25807PlrA (MSMEG_5223) is an essential polar growth regulator in Mycobacterium smegmatisPLOS ONE

Dear Dr. Boutte,

Thank you for submitting your manuscript to PLOS ONE. After careful consideration, we feel that it has merit as pointed out by the two expert reviewers but does not fully meet PLOS ONE’s publication criteria as it currently stands. Therefore, we invite you to submit a revised version of the manuscript that addresses the points raised during the review process.

We look forward to receiving your revised manuscript.

Kind regards,

Delphi Chatterjee

Academic Editor

PLOS ONE

Journal Requirements:

"This work was funded by grant R15GM131317 from the NIH."

 "This work was funded by grant R15GM131317 from the National Institutes of Health to CCB.

https://www.nih.gov

Funders played no role in the study design."

4. Please amend your manuscript to include a reference list. References must be placed at the end of the manuscript and numbered in the order that they appear in the text. For more information on the formatting of references, please visit the author guidelines at: http://journals.plos.org/plosone/s/submission-guidelines#loc-reference-style.

Reviewers' comments:

Reviewer's Responses to Questions

**Comments to the Author**

1. Is the manuscript technically sound, and do the data support the conclusions?

Reviewer #1: Yes

Reviewer #2: Partly

2. Has the statistical analysis been performed appropriately and rigorously? 

Reviewer #1: Yes

Reviewer #2: Yes

3. Have the authors made all data underlying the findings in their manuscript fully available?

Reviewer #1: Yes

Reviewer #2: No

4. Is the manuscript presented in an intelligible fashion and written in standard English?

Reviewer #1: Yes

Reviewer #2: Yes

5. Review Comments to the Author

Reviewer #1: The paper by Quintanilla et al. presents a preliminary characterization of the PrlA protein as a novel factor modulating Wag31 localization to the cell poles. The authors demonstrate that the PrlA protein is essential for viability, and depletion results in aberrant cell wall synthesis. This is a very nice and straight-forward paper. The conclusions are mostly supported by the data (with the minor exception detailed below) and I appreciate the authors’ careful discussion of experimental caveats and biological interpretations.

Detailed Comments:

1. The writing style is a little unusual. Conventionally, new results are presented in past tense, while published data are discussed in present tense. The authors are a little inconsistent with this. There are also a few typos, please read carefully and correct.

2. Line 11 – I do not follow the conclusion that PrlA is involved in restricting PG metabolism to the poles, since they present no evidence of this. Rather, PrlA seems to support Wag31 polar localization (but is not essential for it), so the conclusion is a bit too strong.

3. Line 43, perhaps replace “enzymes” with “synthases” to make less vague.

4. Section in line 112 and following: According to the images and the length distribution of PrlA localization, there seems to be a pronounced midcell localization of PrlA. Can you comment on that?

5. The L5 experiment is very elegant. Just to enhance readability, please add “see methods for details” to line 165, otherwise the reader expects a further explanation of how this works.

6. Western Blots in Fig 3: I can only see a band for the ∆NT lane (not WT), so please supply a more high resolution picture. These data should also be quantified with densitometry, to support the conclusion that the C-terminal truncation is less stable (not obvious from the images). It would also help to point out with an arrow in the figure itself where the expected PrlA band is. Lastly, the PrlA depletion strain could be used as a negative control here.

Reviewer #2: The authors present an interesting study which identifies the PlrA protein having a role in cell division in Mycobacterium smegmatis. The paper is well presented and logical, but there are some areas where key information is missing.

Major points

1. A more detailed description of how the strains were constructed and characterized is needed. Since the identity of strains is critical for understanding the results, the authors should describe in the methods how each strain was constructed. For example, for the inducible strain, what was the deletion made in the chromosome, was this marked, how much was deleted etc. Further details are needed for the fluorescent-tagged and merodiploid strains as well. There was no description of how the strains were characterized or confirmed.

2. The majority of the microscopy panels used images of single cells stitched together. It is not obvious why this was done or what images were excluded. The authors should provide the raw images from which images were derived as supplementary data.

3. A description of each plasmid should be provided in the table. Primers were provided in a table but there was no description of how they were used to construct plasmids. Construction of plasmids should be provided in the methods section.

4. The authors claim that PlrA must be a regulator of Wag31 because it does not have an identifiable enzyme domain. This is overstatement and assumes that all enzyme domains and functions are known. There was no data to support their conclusion as stated.

5. In the survival experiment, the authors did not see any growth as evidenced by an increase in CFU for the “control”. Can they explain this observation?

6. The legend for panel 1C was not sufficient to understand the data presented.

7. L106 The claim that most HADA signal comes from peptidoglycan remodelling is not supported by any data in this paper and this could differ when modulation expressing of PlrA. What is the evidence for this statement in the recombinant strains?

8. There was no characterization of the ATc inducible strain to show similar levels of expression of PlrA as in the wild type when ATc was present. Since this was used as the control for several experiments it is important to know the relative levels in both induced and uninduced conditions.

9. For the L5 switched alleles, the authors should include more characterization of the strains including number and type of transformants and how were the strains confirmed. It was not clear how they could repress PlrA expression from a second integrated vector, this should be explicitly described.

10. For the analysis in Figure 3 - lines 138-144, the authors state that PlrA is not found in other bacteria, so what are they comparing it to to determine conservation of residues?

11. L176 - the authors need to discuss why they see no PlrA-delCT when the strains are viable.

12. For the Wag31 localization studies, there was no wild type strain for comparison. Are the authors sure that their inducible strain is equivalent to the wild type?

13. Lines 263-271 There are some copy/paste errors which make it difficult to understand.

14. L316 - what is the Gfp-mut? More information on the labelled strains is required either in the methods or results section to explain what was constructed and characterized.

15. There was no bibliography in the submitted version.

6. PLOS authors have the option to publish the peer review history of their article (what does this mean?). If published, this will include your full peer review and any attached files.

Reviewer #1: No

Reviewer #2: No

---

## [Author Response · Author response to Decision Letter 0]

27 Nov 2022

Response to reviewer comments

Reviewer #1: The paper by Quintanilla et al. presents a preliminary characterization of the PrlA protein as a novel factor modulating Wag31 localization to the cell poles. The authors demonstrate that the PrlA protein is essential for viability, and depletion results in aberrant cell wall synthesis. This is a very nice and straight-forward paper. The conclusions are mostly supported by the data (with the minor exception detailed below) and I appreciate the authors’ careful discussion of experimental caveats and biological interpretations.

Detailed Comments:

1. The writing style is a little unusual. Conventionally, new results are presented in past tense, while published data are discussed in present tense. The authors are a little inconsistent with this. There are also a few typos, please read carefully and correct.

Corrections made. 

2. Line 11 – I do not follow the conclusion that PrlA is involved in restricting PG metabolism to the poles, since they present no evidence of this. Rather, PrlA seems to support Wag31 polar localization (but is not essential for it), so the conclusion is a bit too strong.

We have added additional text and rewritten parts of the results section to make our conclusion more clear. Fig 1B and C show that peptidoglycan metabolism – as measured by fluorescent D-amino acid staining – becomes localized at the lateral walls instead of the poles upon PlrA depletion. 

3. Line 43, perhaps replace “enzymes” with “synthases” to make less vague.

Change made

4. Section in line 112 and following: According to the images and the length distribution of PrlA localization, there seems to be a pronounced midcell localization of PrlA. Can you comment on that?

Thank you for this suggestion. We added a paragraph to the results to discuss the septal localization of PlrA. 

5. The L5 experiment is very elegant. Just to enhance readability, please add “see methods for details” to line 165, otherwise the reader expects a further explanation of how this works.

Change made. 

6. Western Blots in Fig 3: I can only see a band for the ∆NT lane (not WT), so please supply a more high resolution picture. These data should also be quantified with densitometry, to support the conclusion that the C-terminal truncation is less stable (not obvious from the images). It would also help to point out with an arrow in the figure itself where the expected PrlA band is. Lastly, the PrlA depletion strain could be used as a negative control here.

The Ptet::plrA depletion strain does not have a strep tag on PlrA, so it could not be used as a control here. The wild-type PlrA is serving as a control. We made the requested changes to the figure. 

Reviewer #2: The authors present an interesting study which identifies the PlrA protein having a role in cell division in Mycobacterium smegmatis. The paper is well presented and logical, but there are some areas where key information is missing.

Major points

1. A more detailed description of how the strains were constructed and characterized is needed. Since the identity of strains is critical for understanding the results, the authors should describe in the methods how each strain was constructed. For example, for the inducible strain, what was the deletion made in the chromosome, was this marked, how much was deleted etc. Further details are needed for the fluorescent-tagged and merodiploid strains as well. There was no description of how the strains were characterized or confirmed.

Additional details added. 

2. The majority of the microscopy panels used images of single cells stitched together. It is not obvious why this was done or what images were excluded. The authors should provide the raw images from which images were derived as supplementary data.

It is normal in mycobacterial papers to crop images together to show a representative collection of cells, because mycobacteria clump and don’t tend to spread out neatly on the microscope slides like gram negative and gram positive bacteria. The standard way to indicate cropping is to draw white lines around all the cropped images, which we neglected to do here. We have corrected this by adding the white lines to indicate the crops. As the original images comprise dozens of images, we do not think they add value to the supplemental materials. 

3. A description of each plasmid should be provided in the table. Primers were provided in a table but there was no description of how they were used to construct plasmids. Construction of plasmids should be provided in the methods section.

We added a column to the plasmid table with more information about each plasmid. We added additional information to the methods about the plasmids. 

4. The authors claim that PlrA must be a regulator of Wag31 because it does not have an identifiable enzyme domain. This is overstatement and assumes that all enzyme domains and functions are known. There was no data to support their conclusion as stated.

We changed the wording in lines 84-85 to avoid the unsubstantiated assumption.

5. In the survival experiment, the authors did not see any growth as evidenced by an increase in CFU for the “control”. Can they explain this observation?

We added a paragraph to the results section to explain better how this experiment was done. 

6. The legend for panel 1C was not sufficient to understand the data presented.

Additional explanations added.

7. L106 The claim that most HADA signal comes from peptidoglycan remodelling is not supported by any data in this paper and this could differ when modulation expressing of PlrA. What is the evidence for this statement in the recombinant strains?

We re-wrote this section to make it more clear which conclusions come from previous literature, and which are from this work. 

8. There was no characterization of the ATc inducible strain to show similar levels of expression of PlrA as in the wild type when ATc was present. Since this was used as the control for several experiments it is important to know the relative levels in both induced and uninduced conditions.

We performed a growth curve with the Msmeg wild-type strain and the Ptet::plrA strain at a range of different induction conditions to determine whether the expression level of PlrA affects growth rate. We include these data as in inset into a modified Fig. 1A. The data show that in a range of different PlrA induction conditions, Ptet::plrA cells grow at the same rate as the WT strain. This validates the Atc inducible strain. We included mention of this additional data in the results section. 

9. For the L5 switched alleles, the authors should include more characterization of the strains including number and type of transformants and how were the strains confirmed. It was not clear how they could repress PlrA expression from a second integrated vector, this should be explicitly described.

We added more information in the results to explain this method, and added a note that further details are in the Materials and Methods. 

10. For the analysis in Figure 3 - lines 138-144, the authors state that PlrA is not found in other bacteria, so what are they comparing it to to determine conservation of residues?

We clarified the language here. The gene is present in other bacteria, but has not been experimentally characterized.

11. L176 - the authors need to discuss why they see no PlrA-delCT when the strains are viable.

We added more discussion of this point.

12. For the Wag31 localization studies, there was no wild type strain for comparison. Are the authors sure that their inducible strain is equivalent to the wild type?

As mentioned above, we did a growth curve to address this issue. See inset of revised fig. 1A. 

13. Lines 263-271 There are some copy/paste errors which make it difficult to understand.

Change made.

14. L316 - what is the Gfp-mut? More information on the labelled strains is required either in the methods or results section to explain what was constructed and characterized.

We added more information in the microscopy and strain construction sections of the materials and methods, as well as additions to the plasmid table in the supplement. 

15. There was no bibliography in the submitted version.

Corrected.

---

## [Editor Report · Decision Letter 1]

27 Dec 2022

PlrA (MSMEG_5223) is an essential polar growth regulator in Mycobacterium smegmatis

PONE-D-22-25807R1

Dear Dr. Boutte,

We’re pleased to inform you that your manuscript has been judged scientifically suitable for publication and will be formally accepted for publication once it meets all outstanding technical requirements.

Kind regards,

Delphi Chatterjee

Academic Editor

PLOS ONE
---

## [Editor Report · Acceptance letter]

4 Jan 2023

PONE-D-22-25807R1 

PlrA (MSMEG_5223) is an essential polar growth regulator in *Mycobacterium smegmatis*

Dear Dr. Boutte:

I'm pleased to inform you that your manuscript has been deemed suitable for publication in PLOS ONE. Congratulations! Your manuscript is now with our production department. 

Kind regards, 

on behalf of

Dr. Delphi Chatterjee 

Academic Editor

PLOS ONE